# Fatty Acid Composition and Oxidative Stability of the Lipid Fraction of Skin-On and Skinless Fillets of Prussian Carp (*Carassius gibelio*)

**DOI:** 10.3390/ani10050778

**Published:** 2020-04-30

**Authors:** Piotr Skałecki, Agnieszka Kaliniak-Dziura, Piotr Domaradzki, Mariusz Florek, Monika Kępka

**Affiliations:** Institute of Quality Assessment and Processing of Animal Products, Faculty of Animal Sciences and Bioeconomy, University of Life Sciences in Lublin, Akademicka 13, 20-950 Lublin, Poland; piotr.skalecki@up.lublin.pl (P.S.); mariusz.florek@up.lublin.pl (M.F.); monika.01@poczta.fm (M.K.)

**Keywords:** Prussian Carp, fatty acids, oxidative stability, lipid hydrolysis

## Abstract

**Simple Summary:**

Examination of the proximate nutritional composition of Prussian Carp fish fillets revealed substances with health-promoting or functional properties for humans. High-value constituents, such as eicosapentaenoic (EPA) and docosahexaenoic (DHA) acids, were higher in skin-on fillets than in skinned products, but nutritionally controversial compounds were higher in the former. Fortunately, the obtained values did not exceed the limits adopted for fresh fish and/or fish fit for consumption. This study implies that Prussian Carp can be used on a larger scale in the food industry, thereby facilitating control of the growing domestic population of this species.

**Abstract:**

The aim of the study was to compare the fatty acid profile and content and the oxidative stability of the lipid fraction of Prussian Carp (*Carassius gibelio*) fillets with and without skin. Carp specimens were obtained in spring from a fish farm located in the Lublin Voivodeship. The research material consisted of skin-on (n = 12) and skinless (n = 12) fillets (hand-filleted). Their chemical composition (water, ash, protein, and fat content), calorific value, and nutritional quality index (NQI) were analysed, as well as their fatty acid content and profile. The oxidative stability of the lipids was assessed on the basis of the peroxide value (PV), thiobarbituric acid reactive substances (TBARS) value, and the content of the conjugated dienes and trienes (CD and CT), while determination of the degree of fat hydrolysis was based on the acid value (AV) and free fatty acids (FFA). The Carp fillet with skin contained significantly (*p* ≤ 0.01) more fat (by 2.69 pp) and calories (by 92.16 kJ∙100 g^−1^) than the skinless fillet, and over three times as much n-3 polyunsaturated fatty acids (n-3 PUFA), including eicosapentaenoic (EPA) and docosahexaenoic (DHA) acids. In the skin-on fillet, the lipid oxidation and hydrolysis parameters were significantly higher, but did not deviate from the values specified for fresh fish and/or fish fit for consumption.

## 1. Introduction

Meat of fish is regarded mainly by consumers as a source of biologically active substances, but by-products derived from fish also have high nutritional value. The skin of fish, compared to meat, contains large amounts of protein, including collagen, and iodine. It should be added that under the skin the dark muscles are located, which are rich in fat and other substances, including vitamins [1,2], and which usually are removed along with the skin during skinning.

Prussian Carp (*Carassius gibelio*) is considered one of the main species (alongside Carp) reared in pond farming in Poland. The populations of these fish in open waters are growing. The literature indicates that *C. gibelio* has recently surpassed the native Crucian Carp (*Carassius carassius*) in terms of number of sites [3,4]. This may be due to its hardiness to withstand difficult habitat conditions [5,6]. In Poland, this species is considered expansive and a threat to some fish populations, as in other countries as well [7].

The high content of polyene acids in fish lipids makes them susceptible to oxidation and the appearance of nutritionally controversial compounds, in the form of oxidation and hydrolysis products. Given the documented harmfulness of lipid oxidation products, it is extremely important to objectively assess their content, especially in fish as a food that is highly susceptible to oxidation.

The aim of the study was to compare the fatty acid profile and oxidative stability of the lipid fraction of Prussian Carp (*C. gibelio*) fillets with and without skin.

## 2. Materials and Methods

The study was conducted on 12 Prussian Carp (*C. gibelio*) from a commercial fish farm located in the Lublin Voivodeship (Poland) in spring. All commercial farming activities, including slaughtering, were carried out according to the Code of Good Fishing Practices and under constant supervision of official veterinary inspection. Fish after killing were immediately transported under refrigeration. Killed fish reached the laboratory 2 h after capture. The average body weight and total length of the fish were 783.43 ± 14.45 g and 33.42 ± 1.30 cm. Two types of fillets were obtained from each fish: skin-on (n = 12) and skinless (n = 12). Representative samples (in duplicate) were taken from each fillet.

The chemical composition was determined by reference methods (moisture according to PN-ISO 1442:2000; ash—PN-ISO 936:2000; protein—PN-A-04018:1975/Az3:2002; and fat—PN-ISO 1444:2000). The energy value of 100 g of fillet (kJ·100 g^−1^) was calculated according to Jeszka [8] and the nutritional quality index (NQI) according to Hansen et al. [9], adopting reference values for energy and nutrient intake [10].

Following fat extraction [11], the proportions of fatty acids were determined according to PN-EN ISO 12966-2:2017-05 and PN-EN ISO 12966-1:2015-01. Fatty acid methyl esters were separated by gas chromatography using a Varian GC 3900 gas chromatograph (Walnut Creek, CA, USA).

The acid value (AV), free fatty acids (FFA) and peroxide value (PV) were determined by the method of Koniecko [12], as modified by Joseph et al. [13]. Thiobarbituric acid reactive substances (TBARS) were determined according to Witte et al. [14] and the conjugated dienes (CD) and trienes (CT) were determined by the method of Pegg [15].

Statistical analysis of the results was performed in Statistica 13 (TIBCO Software Inc., Palo Alto, CA, USA). The significance of the differences between the means was verified by a Student’s t-test for independent samples at *p* ≤ 0.05 and *p* ≤ 0.01. In the tables the mean values and standard deviations (SD) are presented.

## 3. Results

Removal of the skin from Prussian Carp fillets significantly increased (*p* ≤ 0.01) the protein content and the higher NQI for protein, but at the same time it significantly (*p* ≤ 0.01) reduced the fat content, NQI for fat, and calorific value (Table 1).

The presence of skin significantly influenced the fatty acid profile of Prussian Carp fillets (Table 2, Appendix A). The skin-on fillet contained significantly (*p* ≤ 0.01) fewer saturated fatty acids (SFA) and more unsaturated fatty acids (UFA), including monounsaturated fatty acids (MUFA), and the ratio of polyunsaturated to saturated fatty acids (PUFA/SFA) was significantly (*p* ≤ 0.01) higher than in the skinless fillet. There was no significant difference in the total share of n-3 acids, while the total share of the n-6 acids was significantly (*p* ≤ 0.01) higher in the skinless fillet, which also had a significantly (*p* ≤ 0.01) lower ratio of total n-3/n-6 acids. The share of DHA was significantly (*p* ≤ 0.01) higher in the fillet without skin, while the skin-on fillet had a higher proportion of EPA and a higher EPA/DHA ratio. The skinless fillet had a significantly (*p* ≤ 0.01) higher total share of saturated branched-chain fatty acids (BCFA), while the share of trans acids (TFA) was the same in both types of fillets.

Presenting the content of fatty acids (especially those with health-promoting effects) in absolute values seems important from the point-of-view of consumers (Table 2 , Appendix A). The skin-on fillets in 100 g contained significantly (*p* ≤ 0.01) more of all the identified fatty acids than the skinless fillets, including over three times the content of valuable PUFA, n-3, EPA, and DHA, which was linked to the fact that the lipid content was three times as high (Table 1).

Table 3 presents the results of the determination of oxidation and lipid hydrolysis products in Prussian Carp fillets. The fillet with skin had a significantly (*p* ≤ 0.01) higher acid value and free fatty acids content than the skinless fillet and a significantly higher (*p* ≤ 0.05) peroxide value and TBARS value (*p* ≤ 0.01). Similar results were obtained for conjugated dienes (*p* ≤ 0.01) and trienes (*p* ≤ 0.05), with significantly higher levels found in the skin-on fillet.

## 4. Conclusions

Prussian Carp fillet with skin contained significantly more fat (by 2.69 pp) and calories (by 92.16 kJ∙100 g^−1^) than the skinless fillet, and also contained significantly more unsaturated fatty acids, while the share of saturated acids was significantly smaller and the n-3/n-6 was significantly more favourable. Due to the higher fat content, the skin-on fillet provided over three times more PUFA fatty acids, n-3 acids and combined EPA and DHA. Taking into account those results it is appropriate to suggest that it is more beneficial for humans to consume *C. gibelio* with skin. The higher lipid content in the skin-on fillet probably determined the susceptibility of lipids to oxidation and hydrolysis. It should be emphasized, however, that the values obtained for indicators did not exceed limits given in the literature [16,17,18,19] for fresh fish and/or fish fit for consumption. The results of the study indicate that Prussian Carp can be used on a larger scale in the food industry, thereby facilitating control of the growing domestic population of this species. Fish skin and the dark muscles have a much higher content of pro-oxidative compounds, which significantly reduces their oxidative stability relative to light muscles [1]. Additional comparative studies are necessary to determine if such differences between the skin, dark, and light muscles also occur in *C. gibelio*.

## Figures and Tables

**Table 1 animals-10-00778-t001:** Proximate composition, calorific value, and nutritional quality index (NQI) of the protein and fat of Prussian Carp (*Carassius gibelio*) fillets (mean ± SD).

Component	Skin-On Fillet	Skinless Fillet
Water (%)	76.6 ± 0.05	78.66 ± 0.01
Protein (%)	18.27 ^A^ ± 0.05	18.97 ^B^ ± 0.12
Fat (%)	3.91 ^B^ ± 0.20	1.22 ^A^ ± 0.12
Ash (%)	1.11 ± 0.09	1.15 ± 0.02
Calorific value (kJ·100 g^−1^)	586.09 ^B^ ± 9.23	493.93 ^A^ ± 10.89
NQI protein	5.24 ^A^ ± 0.06	6.42 ^B^ ± 0.05
NQI fat	0.80 ^B^ ± 0.03	0.30 ^A^ ± 0.02

Means in rows with different letters differ statistically significantly: A, B: *p* ≤ 0.01.

**Table 2 animals-10-00778-t002:** Fatty acid profile (% fatty acids) and content (mg·100 g^−1^ fillet) of Prussian Carp (*Carassius gibelio*) fillets (mean ± SD).

Fatty Acids	% Fatty Acids		mg·100 g^−1^ Fillet	
	Skin-On Fillet	Skinless Fillet	Skin-On Fillet	Skinless Fillet
SFA	21.37 ^A^ ± 0.29	22.82 ^B^ ± 0.15	748.87 ^Y^ ± 10.25	227.09 ^X^ ± 1.49
BCFA	3.80 ^A^ ± 0.02	4.79 ^B^ ± 0.06	133.07 ^Y^ ± 0.75	47.72 ^X^ ± 0.61
UFA	74.84 ^B^ ± 0.31	72.39 ^A^ ± 0.19	2623.09 ^Y^ ± 10.76	720.45 ^X^ ± 1.87
MUFA	38.48 ^B^ ± 0.16	35.31 ^A^ ± 0.18	1348.62 ^Y^ ± 5.70	351.47 ^X^ ± 1.83
PUFA	36.36 ± 0.46	37.07 ± 0.37	1274.47 ^Y^ ± 16.26	368.98 ^X^ ± 3.65
PUFA/SFA	1.70 ^b^ ± 0.05	1.62 ^a^ ± 0.03		
n-3	26.02 ± 0.47	26.40 ± 0.42	912.07 ^Y^ ± 16.50	262.75 ^X^ ± 4.17
n-6	6.77 ^A^ ± 0.06	7.96 ^B^ ± 0.06	237.16 ^Y^ ± 2.23	79.20 ^X^ ± 0.58
n-3/n-6	3.85 ^B^ ± 0.05	3.32 ^A^ ± 0.03		
EPA	11.80 ^B^ ± 0.17	10.58 ^A^ ± 0.18	413.69 ^Y^ ± 5.80	105.28 ^X^ ± 1.84
DHA	6.12 ^A^ ± 0.19	8.64 ^B^ ± 0.13	214.35 ^Y^ ± 6.66	86.04 ^X^ ± 1.31
EPA/DHA	1.93 ^B^ ± 0.05	1.22 ^A^ ± 0.00		
TFA	1.06 ± 0.06	1.06 ± 0.06	37.15 ^Y^ ± 2.17	10.58 ^X^ ± 0.59

Mean values in rows with different letters differ statistically significantly: a, b: *p* ≤ 0.05; A, B: *p* ≤ 0.01. Mean values in rows with different letters differ statistically significantly: X, Y: *p* ≤ 0.01. SFA: total saturated fatty acids; BCFA: total branched-chain fatty acids; UFA: total unsaturated fatty acids; MUFA: total monounsaturated fatty acids; PUFA: total polyunsaturated fatty acids; total n-6 fatty acids; total n-3 fatty acids; EPA: eicosapentaenoic acid (C20:5n-3); DHA: docosahexaenoic acid (C22:6n-3); TFA: total trans fatty acids.

**Table 3 animals-10-00778-t003:** Acid value (AV), free fatty acids (FFA), peroxide value (PV), thiobarbituric acid reactive substances (TBARS), and content of conjugated dienes (CD) and trienes (CT) of Prussian Carp (*Carassius gibelio*) fillets (mean ± SD).

Parameter	Skin-On Fillet	Skinless Fillet
AV (mg KOH/g fat)	5.06 ^B^ ± 0.46	2.93 ^A^ ± 0.16
FFA (%)	2.54 ^B^ ± 0.23	1.47 ^A^ ± 0.08
PV (mEq O_2_/kg fat)	3.69 ^B^ ± 0.41	1.02 ^A^ ± 0.26
TBARS (mg MDA/kg meat)	6.01 ^B^ ± 0.35	2.52 ^A^ ± 0.16
CD (mmol of hydroperoxides/kg of oil)	32.80 ^B^ ± 1.36	27.88 ^A^ ± 0.08
CT (mmol of hydroperoxides/kg of oil)	5.29 ^b^ ± 0.27	4.74 ^a^ ± 0.06

Mean values in rows with different letters differ statistically significantly: a, b: *p* ≤ 0.05; A, B: *p* ≤ 0.01. KOH: potassium hydroxide; mEq: milliequivalent; MDA: malondialdehyde.

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
