# Peer review of "Fatty Acid Composition and Oxidative Stability of the Lipid Fraction of Skin-On and Skinless Fillets of Prussian Carp (Carassius gibelio)"

_animals, 2020, doi:10.3390/ani10050778_

Round 1
Reviewer 1 Report
I agree to publish this manuscript as a communication in this journal.
Reviewer 2 Report
Most of my suggestions were accepted by the authors. I have no comments on the revised MS.
Reviewer 3 Report
The aim of the study was to compare the fatty acid profile and content and the oxidative stability of the lipid fraction of Prussian Carp (Carassius gibelio) fillets with and without skin. About the experimental design, how about compare the properties among skin, dark meat and white meat? It might be simple to achieve the above objective.
Round 2
Reviewer 3 Report
Please change Saturated acid to Saturated fatty acid.
As the skin-on fillet contained significantly (P ≤ 0.01) fewer saturated acids (SFA) and more unsaturated acids (UFA), including monounsaturated fatty acids (MUFA), and the ratio of polyunsaturated to saturated fatty acids (PUFA/SFA) was significantly (P ≤ 0.01) higher than in the skinless fillet. maybe we can encourage the consumer to eat more fish with skin-on fillet?
Author Response
Please see the attachment.

This manuscript is a resubmission of an earlier submission. The following is a list of the peer review reports and author responses from that submission.
Round 1
Reviewer 1 Report
The MS described the ratio of fat and valuable PUFA of Carassius gibelio. It can contribute scientific progress by basal information of fisheries materials.
It is descriptive research, therefore, I do not have any idea to brush up. I noted the followings for readers providing good information:
Line 46: polyunsaturated fatty acids > polyunsaturated fatty acids (PUFA)
Lines 53-54: I have a little complicated. It means “Under the skin fish have dark muscles; skin is rich in fat and other substances, including vitamins” ?
Line 61: You use “crucian carp” for C. carassius and “Prussian carp” for C. gibelio. I think all common name should be wrote as proper noun. Therefore, Crucian Carp and Prussian Carp are better for using in English sentences.
Line 87: I could not find TBARS from previous sentences. You should explain the abbreviation.
Line 178: Is it not needed that information of seasonal influences for several substances in comarison?
Reviewer 2 Report
This paper provides a comparison of crucian carp fillet based on food chemistry. In my opinion, it would be appropriate to submit this paper to a specialized journal in food science or food chemistry.
The crucian carp used in this study is a cultured fish. The fatty acid component of fish is affected by its diet. Would the authors be able to provide data on the fatty acid composition of the diet of this cultured crucian carp used in the study, if possible?
In this paper, the comparison of the components of the two fillets is based solely on component analysis at the time of transfer to the laboratory.
It is considered that there are few data to compare oxidative stability.
Is it possible to add the following data?
What state and how long did the crucian carp fillet reach the lab after it was landed?
In order to compare the oxidative stability, it is necessary to measure the components immediately after landing, and to examine the change in components during storage or transportation.
The authors analyzed the acid value. Can the authors add data on lipid composition such as polar lipids and free fatty acids?
I think it would be interesting to have data comparing with marine fish.
Round 2
Reviewer 2 Report
I accept the author's response.